# THWARTING FINITE DIFFERENCE ADVERSARIAL ATTACKS WITH OUTPUT RANDOMIZATION

## ABSTRACT

Adversarial input poses a critical problem to deep neural networks (DNN). This problem is more severe in the "black box" setting where an adversary only needs to repeatedly query a DNN to estimate the gradients required to create adversarial examples. Current defense techniques against attacks in this setting are not effective. Thus, in this paper, we present a novel defense technique based on randomization applied to a DNN's output layer. While effective as a defense technique, this approach introduces a trade off between accuracy and robustness. We show that for certain types of randomization, we can bound the probability of introducing errors by carefully setting distributional parameters. For the particular case of finite difference black box attacks, we quantify the error introduced by the defense in the finite difference estimate of the gradient. Lastly, we show empirically that the defense can thwart three **adaptive** black box adversarial attack algorithms.

## 1   INTRODUCTION

The success of deep neural networks has led to scrutiny of the security vulnerabilities in deep neural network based models. One particular area of concern is weakness to adversarial input: carefully crafted inputs that resist detection and can cause arbitrary errors in the model (1; 2). This is especially highlighted in the domain of image classification, where an adversary creates an image that resembles a natural image to a human observer but easily fools deep neural network based image classifiers (3).

Different types of adversarial attacks exists throughout the lifecycle of a deep neural network model. For example, adversaries can attack a model during training by injecting corrupting data into the training set or by modifying data in the training set. However, inference time attacks are more worrisome as they represent the bulk of realistic attack surfaces (4; 5; 6).

The input created under an inference time attack is known as an adversarial example and methods for generating such examples have recently attracted much attention. In most cases, the adversarial example is created by perturbing a (correctly classified) input such that the model commits an error. The perturbation applied to the "clean" input is typically constrained to be undetectably small (7; 8; 9).

Defending againsts adversarial attacks on deep neural networks is crucial and has seen relatively slow progress compared to the sophistication and progress of adversarial attacks (10; 11; 12; 13; 14; 15). This is because it is difficult to prove a defense can withstand the types of attacks it is designed for especially when the defense must prove capable of withstanding "adaptive" adversaries that have knowledge of the details of the defense (16).

A defense against adversarial attacks is defined by the threat model it is designed to defend against (16). The most permissive threat model makes the weakest assumptions about the adversary. One such threat model can be assuming the adversary has complete knowledge of the model, including architecture and parameters of the underlying network. This is known as the "white-box" setting. Other threat models can also be useful, such as assuming the adversary has knowledge of the network architecture but not the parameters of the model. More restrictive threat models allow only so called "black-box" attacks, attacks that can create adversarial examples without having access to the model architecture or weights and only accessing some form of output of the model.

The rest of the paper is organized as follows. In Section 2, we discuss the threat model considered. In Section 3, we describe valid black box attacks under the threat model. Output randomization as a

defense is described in Section 4. We show empirical results in Section 5, cover related approaches in Section 6, and conclude in Section 7.

## 2  THREAT MODEL

**Adversary goals:** The goal of the adversary is to force the classifier to commit an error within a distortion limit, such that the example crafted by the adversary is similar to the original example. The adversary induces such an error in an untargeted or targeted attack. The goal of an untargeted attack is misclassification of an input, whereas the goal of a targeted attack is misclassification of an input as a class specified by the adversary.

**Adversary knowledge:** The adversary has access to the model only at the input and output level, and has no knowledge of its architecture or parameters. This black-box adversary is aware of the details of the defenses protecting the model and the type of randomness associated with any defense but not the exact random numbers generated.

**Adversary capability:** The adversary only has access to the model by providing examples as input and observing the output probability vector generated by the model as output. The adversary can modify aspects of the input in any way as long as it remains similar to the original input. This similarity is controlled by an $l_p$ distortion penalty on the adversarial image. Common choices for the distortion penalty include $l_1, l_2$ and $l_\infty$. We use the $l_2$ perturbation penalty as this type of attack results in the strongest attacks (17). Attackers are allowed to query the model up to a maximum limit, which we increase to our computational limit to strengthen attacks. In addition to a maximum query limit, attackers often use an early stopping parameter to avoid wasting computation on unpromising direction during optimization. We also increase this parameter to test our defense against stronger attacks.

## 3  EXISTING BLACK BOX ATTACKS

Black box attacks are called "gradient-free" attacks since they do not involve computing gradients of the input by backpropagation on the model under attack. Instead the gradients of the input are estimated by using the finite difference estimate for each input feature.

Black box attacks can be categorized by the type of output the adversary is allowed to observe: logit layer, class probabilities/softmax layer, or top $k$ class probabilities/labels.

In general, designing successful black box attacks in the label only setting is much harder than the setting where the logit layer or softmax layer is available to the attacker. We consider the easiest setting for black box attacks, where all the class probabilities are available.

### 3.1  ZOO BLACK BOX ATTACK

The Zeroth Order Optimization based black-box (ZOO) attack (18) is a method for creating adversarial examples for deep neural networks that only requires input and output access to the model. ZOO adopts a similar iterative optimization based approach to adversarial example generation as other successful attacks, such as the Carlini & Wagner (C&W) attack (17). The attack begins with a correctly classified input image $x$, defines an adversarial loss function that scores perturbations $\delta$ applied to the input, and optimizes the adversarial loss function using gradient descent to find $\delta^*$ that creates a succesful adversarial example. Specifically, gradient descent is used to find $\delta^*$ such that:

$$f(x + \delta^*) = y^a$$

$$\|x - (x + \delta^*)\| \le \epsilon$$

Namely, that the perturbed input $x + \delta^*$ successfully fools the classifier $f$ to predict the incorrect class $y^a$ and that the perturbed input is similar to the original input up to some distortion limit $\epsilon$.

The primary (and strongest) adversarial loss used by the ZOO attack for targeted attacks is given by:

$$L(x, t) = \max \left\{ \max_{i \ne t} \{\log f(x)_i - \log f(x)_t\}, -\kappa \right\} \tag{1}$$

Where $x$ is an input image, $t$ is a target class, and $\kappa$ is a tuning parameter. Minimizing this loss function over the input $x$ causes the classifier to predict class $t$ for the optimized input. For untargeted attacks, a similar loss function is used:

$$L(x) = \max\left\{\log f(x)_i - \max_{j \neq i}\left\{\log f(x)_j\right\}, -\kappa\right\} \tag{2}$$

where $i$ is the original label for the input $x$. This loss function simply pushes $x$ to enter a region of misclassification for the classifier $f$.

In order to limit distortion of the original input, the adversarial loss function is combined with a distortion penalty in the full optimization problem. This is given by:

$$\min_x \|x - x_0\|_2^2 + c \cdot L(x, t)$$
$$\text{subject to } x \in [0, 1]^n$$

In the white box setting, attackers can take advantage of the backprogation algorithm to calculate the gradient of the adversarial loss function with respect to the input coordinates ($\frac{\delta L}{\delta x_i}$) and solve the optimization problem using gradient descent. In lieu of this, ZOO uses "zeroth order stochastic coordinate descent" to optimize input on the adversarial loss directly. This is most easily understood as a finite difference estimate of the gradient of the input with the symmetric difference quotient (18):

$$\frac{\delta L}{\delta x_i} \approx g_i := \frac{L(x + he_i) - L(x - he_i)}{2h}$$

with $e_i$ as the basis vector for coordinate/pixel $i$ and $h$ set to a small constant. The ZOO attack uses this approximation to the gradients to create an adversarial example from the given input. Note that for an image with $n$ pixels, computing an estimate of the gradient with respect to each pixel requires $2n$ queries to the model. Since this is usually prohibitive, the ZOO attack circumvents this by only estimating the gradients for a subset of coordinates at each step. ZOO also uses dimensionality reduction and a hierarchical approach to further increase the efficiency of the attack and show empirically that these methods are effective (18).

### 3.2 Query Limited (QL) black box attack

A similar approach to ZOO is adopted by (19) in a query limited setting. Like ZOO, the QL attack estimates the gradients of the adversarial loss using a finite difference based approach. However, the QL attack reduces the number of queries required to estimate the gradients by employing a search distribution. Natural Evolutionary Strategies (NES) (20) is used as a black box to estimate gradients from a limited number of model evaluations. Projected Gradient Descent (PGD) (12) is used to update the adversarial example using the estimated gradients. PGD uses the sign of the estimated gradients to perform an update: $x^t = x^{t-1} - \eta \cdot \text{sign}(g_t)$, with a step size $\eta$ and the estimated gradient $g_t$. The estimated gradient for the QL attack using NES is given by:

$$g_t = \sum_{i=1}^m \frac{L(x + \sigma \cdot u_i) \cdot u_i - L(x - \sigma \cdot u_i) \cdot u_i}{2m\sigma}$$

where $u_i$ is sampled from a standard normal distribution with the same dimension as the input $x$, $\sigma$ is the search variance, and $m$ is the number of samples used to estimate the gradient. The difference between this approach and ZOO is that ZOO attempts to estimate the gradient with respect to one coordinate at a time while this approach averages over perturbations to many coordinates to estimate the entire gradient directly.

In the next section, we show that applying a simple randomization function (that does not affect model accuracy) to the output of a model causes these types of attacks to fail even if the attack is adapted to the specific randomization function.

## 4 Thwarting black box attacks

The intuition behind output randomization is that a model may deliberately make errors in its predictions in order to thwart a potential attacker. This simple idea introduces a tradeoff between accurate predictions and the effectiveness of finite difference based black box adversarial attacks.

Output randomization for a model that produces a probability distribution over class labels replaces the output of the model $p$ by a stochastic function $d(p)$. The function $d$ must satisfy two conditions:

1. The probability of misclassifying an input due to applying $d$ is bounded by $K$
2. The vector $d(p)$ prevents adversaries under the given threat model from generating adversarial examples.

The first condition ensures that the applied defense minimally impacts non-adversarial users of the model, such as users of an online image classification service. The effectiveness of the defense comes from satisfying the second condition as the introduced randomness must prevent an adversary (in the appropiate setting) from producing an adversarial example.

In the following two sections, we consider a simple noise-inducing function $d(p) = p + \epsilon$ where $\epsilon$ is a random variable.

## 4.1 MISSCLASSIFICATION RATE

A simple function useful for defending a model is the gaussian noise function $d(p) = p + \epsilon$ where $\epsilon$ is a gaussian random variable with mean $\mu$ and variance $\sigma^2$ ($\epsilon \sim \mathcal{N}(\mu, \sigma^2 \cdot \mathbf{I}_C)$). In the black box setting, a user querying the model with an input $x$ receives the perturbed vector $d(p)$ instead of the true probability vector $p$. Note that $d(p)$ does not necessarily represent a probability mass function like $p$.

To verify that this function satisfies the first condition above, we wish to know the probability that the class predicted by the undefended model is the same as the class predicted by the defended model. If the output of the model for an input $x$ is $p$, we will refer to the maximum element of $p$ as $p_m$ and the rest of the elements of $p$ in decreasing order as $p_2, p_3, ... p_C$.

Suppose the model correctly classifies the input $x$ in the vector $p$, we can express the probability that $x$ is misclassified in the vector $d(p)$ as:

$$\sum_{i=2}^{C} \mathbb{P}(d(p_i) > d(p_m))$$

We can write $\mathbb{P}(d(p_i) > d(p_m))$ for $i = 2, 3, ... C$ as:

$$\mathbb{P}(d(p_i) > d(p_m)) = \mathbb{P}(p_i + \epsilon_i > p_m + \epsilon_m)$$

If we define $\delta_i = p_m - p_i$, as shown in Figure 1a, and since $e_i := \epsilon_i - \epsilon_m$ is itself a gaussian with mean $\mu_i - \mu_m$ and variance $\sigma_i^2 + \sigma_m^2$ then we can write:

$$\mathbb{P}(e_i > \delta_i) = 1 - \mathbb{P}(e_i \leq \delta_i) = 1 - \mathbb{P}\left(\frac{e_i - \mu_i + \mu_m}{\sigma_i^2 + \sigma_m^2} \leq \frac{\delta_i - \mu_i + \mu_m}{\sigma_i^2 + \sigma_m^2}\right)$$

Using the cumulative distribution function of a standard gaussian distribution $\Phi$, we can write the misclassification probability as:

$$K := \mathbb{P}(d(p_i) > d(p_m)) = 1 - \Phi\left(\frac{\delta_i - \mu_i + \mu_m}{\sigma_i^2 + \sigma_m^2}\right) = \Phi\left(-\frac{\delta_i - \mu_i + \mu_m}{\sigma_i^2 + \sigma_m^2}\right)$$

For the special case of a gaussian noise function $d(p)$ with mean 0 and variance $\sigma^2$ we would like to fix the probability of misclassification to a value $K$ and compute the appropiate variance $\sigma^2$. We can use the inverse of the standard gaussian cdf $\Phi^{-1}$, or the probit function, to write this easily:

$$\sigma^2 = -\frac{\delta_i}{2\Phi^{-1}(K)}$$

Note that the desired misclassification rate $K < 0.5$ in any real case and so the rhs will be positive. If we consider $\delta_i$ as the confidence of the model, then the allowable variance will be larger when the model is confident and smaller otherwise. We show the calculations above for one class $i$, the misclassification probability ($K$) and level of noise ($\sigma^2$) can be set for each class separately. In Figure 1b we show the maximum allowable variance for different misclassification rates.

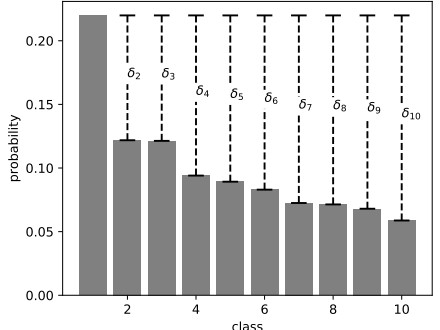 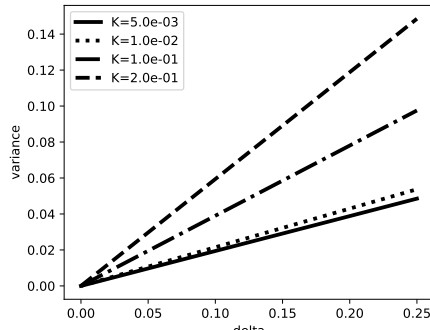

(a) Probability distribution over classes generated by a classification model. $\delta_i$ represents the confidence of the model's prediction.

(b) Maximum output randomization $\sigma^2$ vs $\delta_i$ for misclassification rates $K = \{20\%, 10\%, 1\%, 0.5\%\}$

Figure 1: Controlling misclassification caused by output randomization

## 4.2 Finite difference gradient error

To verify the function $d(p)$ satisfies the second condition, that it introduces error that prevents a finite difference based black box attack, we show the effect of the output randomization on the gradient accessible to the adversary.

Finite difference (FD) based approaches involve evaluating the adversarial loss at two points, $x + he_i$ and $x - he_i$, close to $x$ (with small $h$ and unit vector $e_i$) and using the slope to estimate the gradient of the loss with respect to pixel $i$ of the input. For a loss function $L$, the finite difference estimate of the gradient of pixel $i$ is given by:

$$g_i = \frac{L(f(x + he_i)) - L(f(x - he_i))}{2h}$$

Here, we write the adversarial loss function (either the untargeted loss in Equation 2 or the targeted loss in Equation 1) in terms of the output of the model to make explicit the dependence of $L$ on the output vector of the network $p = f(\cdot)$. $p$ and $p'$ are used to distinguish between the two output vectors needed to compute the gradient estimate. When the network is defended using output randomization, the function $d()$ is applied to the output vector of the network. Thus, the finite difference gradient computed by the attacker is:

$$\gamma_i = \frac{L(d(p)) - L(d(p'))}{2h}$$

The error in the FD gradient introduced by the defense is given by $|g_i - \gamma_i|$. When $d$ is a function that adds noise $\epsilon$ to the output of the network, the expected value of the error is:

$$|E[g_i - \gamma_i]| = \left| g_i - E\left[\frac{L(p + \epsilon) - L(p' + \epsilon')}{2h}\right]\right|$$

This error term depends on the choice of the adversarial loss function $L(\cdot)$. Since the untargeted attack is generally considered easier than the targeted attack, consider how the gradient error of the defended model behaves under the untargeted adversarial loss function. For untargeted attacks, we simplify the loss function to: $L_u(p) = \log(p_c) - \log(p_o) = \log(\frac{p_c}{p_o})$ where $p_c$ is the probability of the true class and $p_o$ is the maximum probability assigned to a class other than the true class of the input image.

Substituting the untargeted adversarial loss for $L(\cdot)$ we see:

$$|E[g_i - \gamma_i]| = \left| g_i - \frac{1}{2h}E\left[\log(\frac{p_c + \epsilon_c}{p_o + \epsilon_o}) - \log(\frac{p'_c + \epsilon'_c}{p'_o + \epsilon'_o})\right]\right|$$

We use a second order Taylor series approximation of $E[\log(X)] \approx \log(E[X]) - \frac{Var[X]}{2E[X]^2}$ to approximate the expectations. If we further assume $\epsilon$ is zero-mean with variance $\sigma^2$, then $E[p + \epsilon] = p$ and the expectation of the defended gradient is approximately:

$$|E[g_i - \gamma_i]| \approx \left| \frac{\sigma^2}{4h} \left( \frac{\sigma^2 + p_o^2 + p_c^{2'}}{p_c^{2'} p_o^2} - \frac{\sigma^2 + p_o^{2'} + p_c^2}{p_c^2 p_o^{2'}} \right) \right|$$

This approximation summarizes the suprising effect output randomization has on finite difference based black box attacks. Firstly, it is easy to see that the error scales with the variance of $\epsilon$ (in the zero mean case). Even when the adversary adapts to the defense by averaging over the output randomization the variance is only reduced linearly by the number of samples. Secondly, even in expectation the error is never non-zero. This is because one of two cases must be true in order for the error to be zero:

1. $p_c == p_o$ and $p'_c == p'_o$
2. $p_c == p'_c$ and $p_o == p'_o$

Case 1 cannot occur because it implies the model is predicting two different classes simultaneously. Case 2 will only occur if $L(p) == L(p')$ which means $g_i = 0$.

The reason for this behavior is mostly due to the $\log$ operation in the adversarial loss function $L$. As noted by the authors in (18), the $\log$ is crucial to the success of finite difference black box attacks as well trained models yield skewed distributions in the output $p = f(x)$. In our experiments, we show that this behavior holds in real world experiments on image classification datasets.

## 5 Empirical results

To evaluate the output randomization defense against black box attacks, we select three successful black box attacks (ZOO (18), QL (19), and BAND (21)) on benchmark image classification datasets (MNIST (22), CIFAR10 (23), and ImageNet (24)). For all defended models, we use $\epsilon \sim \mathcal{N}(0, \sigma^2 \cdot \mathbf{I}_C)$. Details of both the attacks and defenses can be found in our code [1] and the appendix. We follow the guidelines laid out in (16), most importantly we attempt to adapt the attacks to the proposed defense. In this case, this means the attacker is aware of the type of randomization applied to the output of the network. Therefore, we adapt the attacks by allowing the attacker to average over the output randomization in an attempt to bypass the defense. This type of adaptation has been shown to overcome certain types of input and model randomization defenses (25). In our results, we refer to this as the *adaptive* attack.

As a sanity check, we also measure the attack success rate of a white box attacker (Carlini & Wagner L2 (17) attack) with randomized output. We find that the defense has some success in defending against this attack in the non-adaptive case. However the adaptive white box attacker is able to overcome the output randomization by averaging over a small number of samples. This is summarized in Figure 2.

Our main set of experiments is shown in Figure 3 and show the effects of output randomization on the non-adaptive and adaptive ZOO attacks. We show that the defense reduces the attack success rate significantly even in the adaptive attacker setting, where we average over randomness and double attack iterations. The effectiveness of the defense was not affected by targeted or untargeted attacks. Table 1 summarizes the results for three finite difference based black box attacks on ImageNet.

It is important to show how a defense affects the "normal" operating properties of a model and this is typically demonstrated by comparing the test set accuracy of the defended model to the undefended model. Figure 2a and Figure 3a show the effect of increasing noise levels on test set accuracy. Output randomization is an effective defense against black box attacks at noise levels as small as $\sigma^2 = $ 1e-4 where model performance is identical to undefended models.

---

[1]Our attack code is based on `https://github.com/IBM/ZOO-Attack`, `https://github.com/labsix/limited-blackbox-attacks`, and `https://github.com/MadryLab/blackbox-bandits` for the ZOO, QL, and BAND attacks respectively.

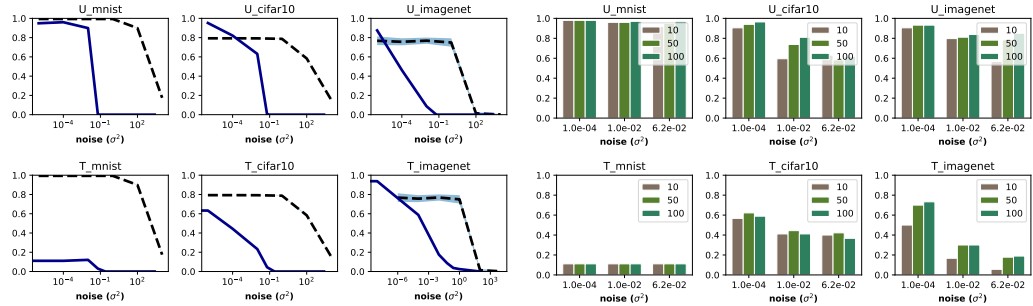

(a) Attack success rate (solid) and test set accuracy (dashed) vs variance for the non-adaptive attacker. Output randomization is not effective against a white box attacker at small noise levels.

(b) Attack success rate vs variance (groups) for adaptive attacker with increasing averaging (10, 50, 100 samples). Averaging allows the white box adaptive attacker to overcome output randomization.

Figure 2: Carlini & Wagner (17) white box attack versus output randomization. Top row shows untargeted attacks, bottom row shows targeted attacks.

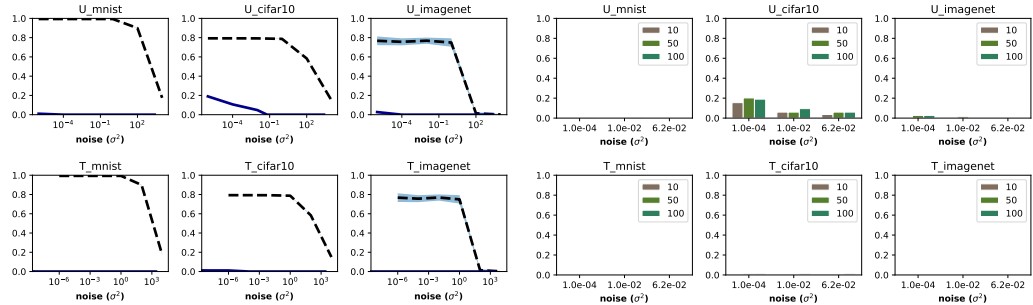

(a) Attack success rate (solid) and test set accuracy (dashed) vs variance for the non-adaptive attacker. Output randomization blocks attacks even at very small noise levels ($\sigma^2 < $ 1e-6).

(b) Attack success rate vs variance (bar groups) for adaptive attacker with increasing averaging (10, 50, 100 samples). Averaging does not improve attack success rate of the black box attacker.

Figure 3: ZOO (18) black box attack versus output randomization. Top row shows untargeted attacks, bottom row shows targeted attacks. Compare this figure with the white box attack results in Figure 2.

Defensive techniques without output randomization are still vulnerable to black box attacks. We evaluated defensive distillation (10) and input randomization (26) against the ZOO attack and found that these defenses did not reduce the attack success rate significantly as shown in Table 2. Distillation is shown to be vulnerable to white box attacks in (17), we show it is also vulnerable to finite difference black box attacks. Input randomization (26) has limited success in defending against ZOO (reducing attack success rate to 0.76), however it is not as effective as output randomization. This is because randomization applied at the input is not guaranteed to affect the finite difference gradient estimates. In addition, input randomization also does not allow fine control over model accuracy.

Table 1: Output randomization vs 3 black box attacks on 100 correctly classified ImageNet examples measured by attack success rate (fraction of examples misclassified)

| Noise Variance | ZOO (18) | QL (19) | BAND (21) |
|---|---|---|---|
| (undefended) | 0.69 | 1.00 | 0.92 |
| 1.00e-4 | 0.03 | 0.73 | 0.58 |
| 1.00e-2 | 0.00 | 0.02 | 0.07 |
| 5.76e-2 | 0.00 | 0.01 | 0.06 |

Table 2: ZOO black box attack success rate vs three defenses

| Dataset | Ex. Type | Distillation(10) | Mitigation(26) | OR (ours) |
|---------|----------|------------------|----------------|-----------|
| MNIST | Targeted | 1.00 | - | **0.00** |
| MNIST | Untargeted | 0.99 | - | **0.01** |
| CIFAR10 | Targeted | 1.00 | - | **0.011** |
| CIFAR10 | Untargeted | 1.00 | - | **0.19** |
| ImageNet | Untargeted | - | 0.76 | **0.005** |

## 6 RELATED WORK

In this section, we discuss related gradient masking or obfuscated gradient defenses. We will focus on proactive defenses, which attempt to make a network robust, compared to reactive defenses, which attempt to detect adversarial examples. (25) defined three ways to obfuscate gradients: shattered gradients, exploding/vanishing gradients, and stochastic gradients.

**Shattered gradients** are a non-differentiable defense that causes a gradient to be nonexistent or incorrect. This can be done unintentionlly by introducing numeric instability. (27) and (28) proposed shattered gradients defenses that introduce a non-differentiable and non-linear discretization to a model's input. These transformation are ineffective as black box attacks are agnostic of input randomization.

**Exploding gradients** make a network hard to train because of an extremely deep neural network. This is generally done by using an output of a neural network as the input of another. (29) and (30) both proposed defenses that utilize GANs. However, (25) shows that these defenses can be bypassed using transferability of adversarial attacks. The transferability property allows an attacker to use adversarial examples created using one model (often a surrogate model) to fool another model (31). Although this is a valid attack vector for even black box models, we do not consider this type of attack in this work.

**Stochastic gradients** randomize gradients by introducing some randomization to the network or randomly transforming the input to the network. (32) proposed a stochastic gradients defense in which a random subset of activations are pruned. (26) introduced randomness by randomly rescaling input images. (25) showed that an adaptive attacker can bypass these defenses by computing the expected value over multiple queries. We test a similar approach on output randomization and show it is not effective in the black box case (Figure 3b).

Although other defense methods consider introducing randomness to the input or model itself, this work is the first to our knowledge to consider randomizing the output of the model directly.

## 7 CONCLUSION

Black box attacks based on finite difference gradient estimates pose a threat to classification models without needing priveleged access to the model. In this paper, we show that this threat can averted by introducing simple types of randomization to the output of the model. Our empirical results show that even very small ($\sigma^2 = $ 1e-6) perturbations to the output can prevent these type of attacks.

Although our work shows an encouraging result for defending against black box attacks, we show that output randomization (or other types of randomization) do not prevent white box attacks (25). Furthermore, output randomization only prevents query based black box attacks and does not address the problem of *transfer* attacks (33; 31). Other attacks, such as $\mathcal{N}$Attack (34), utilize derivative-free optimization to find adversarial examples bypassing the need for finite difference estimates. We leave defense against these types of attacks to future work.

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

APPENDIX

ATTACK DETAILS

For our evaluation, we trained models for MNIST, CIFAR10, and ImageNet that acheived 99%, 79%, and $\sim$ 72% test set accuracies respectively using the code provided by (18). Non-adaptive attacks were conducted using the parameters suggested by the attacks. For ZOO, these include using the ADAM (35) solver, 9 binary search steps, and image resizing + reset ADAM for ImageNet. The adaptive attackers (both white and black) were modeled in two ways. (i) We added averaging over randomness to the attack and (ii) the number of iterations was doubled. For all of our experiments, we averaged the attack success rate over 100 images and report the mean value over 30 runs.

EXTENSIONS

Other randomization functions can be used to introduce randomness in the output instead of gaussian noise. As an extension, we considered noise sampled from a gaussian mixture model with random mixing coefficients and parameters. In theory, this type of randomization should be harder for attackers to average over and avoid. However, we saw that a white box attacker could still average over the added noise with 100 samples and circumvent the defense. Since the simple gaussian noise was effective in the blackbox case, we demonstrate our results using gaussian noise.

We also experimented with randomization of the logit layer and observed improvements over softmax layer randomization. However, we chose to present the more general softmax layer randomization for clarity.

Figure 4 shows the error between the finite difference gradients for the ZOO attack on a defended and undefended model at varying noise levels. As we expect, increased noise levels cause the overall error (measured by the norm of difference of the gradients) to increase dramatically.

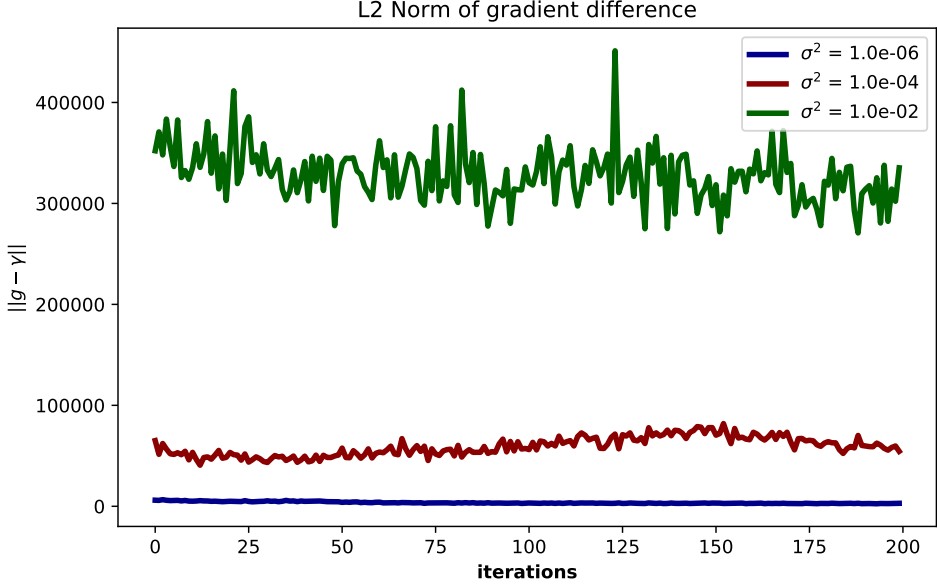

Figure 4: $L_2$ norm of difference between the true finite difference gradient calculated by ZOO on the undefended model and the finite difference gradient calculated on the defended model with increasing variance. As expected the error is significantly higher for noise with larger variance.

