# OpenReview forum: "Thwarting finite difference adversarial attacks with output randomization"
_ICLR.cc/2020/Conference — Reject_

### Official Review · AnonReviewer3 · 2019-10-22
**Official Blind Review #3**

**Rating:** 3

**Review:**

The authors present a randomization defense in the black-box threat model, where bounded l_2 norm perturbations are allowed. In the presented scheme, the defender adds Gaussian noise to every coordinate of the output probability vector before returning an inference result.


The paper suffers from an incomplete evaluation, and therefore I cannot recommend acceptance.


Details:
* Completeness of evaluation. As shown by Ilyas et al [1] and Cheng et al [2] and Brendel et al [3], black-box attacks can succeed even when only information about the label is present. Thus, I suspect that an attacker can simply run any of these attack algorithms in order to fool the model. In addition, if we use enough samples in the NES-based “Query-Limited” algorithm of Ilyas et al (that is, with enough samples per step) then we should be able to perfectly mimic the white-box attack, which as shown in Figure 2 is effective.
* Potential flaw in evaluation. BAND performs worse than QL on the undefended classifier in Table 1, which should not occur. I would check to make sure that the attacks are applied correctly here.
* Lacking details in evaluation. I could not find how many samples are used to perform the attacks in Table 1. It is hard to evaluate the defense without knowing how many samples are used in each algorithm.


[1] https://arxiv.org/abs/1804.08598
[2] https://arxiv.org/abs/1807.04457
[3] https://arxiv.org/abs/1712.04248


**Experience Assessment:**

I have published in this field for several years.

**Review Assessment: Checking Correctness Of Derivations And Theory:**

I assessed the sensibility of the derivations and theory.

**Review Assessment: Checking Correctness Of Experiments:**

I carefully checked the experiments.

**Review Assessment: Thoroughness In Paper Reading:**

I read the paper thoroughly.

---

> ### Author Response · Authors · 2019-11-14
> **Clarification of evaluation**
>
> We reiterate that our proposed method is intended to highlight the weaknesses of finite difference based black box attacks to simple perturbations to the model output (See our response to Reviewer #2 for more details). Indeed, we state this explicitly in the paper in several places in the manuscript in an effort to place our work in the correct context and prevent misunderstandings. Thus, we did not consider the “label-only” black box attacks referenced in the review. We don’t agree with the reviewer’s suspicion that these types of attacks are able to deal with the randomization proposed in our work but leave those experiments to future work.
> -	Concerning the number of samples used by the BAND and QL attacks (and to address the reviewer’s comments about the NES-based attack), we allowed these attacks to use up to a maximum of 10,000 samples which we considered sufficient. We are happy to include these details in the Attack Details section of the appendix.
> -	The BAND and QL performance on the undefended classifier in Table 1 was measured directly using the code provided by the authors. We could not find any flaws in their code, can the reviewer elaborate more on this?

---

> > ### Comment · AnonReviewer3 · 2019-11-14
> > **Threat model**
> >
> > If the black-box threat model is not the intended threat model, what is? Black-box attacks attacks are only interesting from a theoretical or security perspective, and this paper is mostly written from the security perspective---in this case we must have a concrete threat model.
> >
> > Re: more samples --- without using more samples it is hard to tell when the defense breaks down. However, knowing the limitations of a defense is important from both a science perspective and a practical perspective.
> >
> > Re: applying algorithms correctly---there are always a number of caveats when applying algorithms from another author directly (e.g. making sure that the images are scaled properly, tuning critical hyperparameters). It would be good to check this over and ensure that you have searched over the parameters properly.

---

> > > ### Author Response · Authors · 2019-11-15
> > > **Adversary**
> > >
> > > To be clear, we consider an adversary that mounts a finite-difference black box attack with the goals, knowledge, and capabilities laid out in section 2. Our main result is showing the brittleness of these types of attacks specifically their susceptibility to randomization of the model outputs.
> > >
> > > Thanks for your comments!

---

### Official Review · AnonReviewer1 · 2019-10-25
**Official Blind Review #1**

**Rating:** 6

**Review:**

This paper presents a method for defending black box (in particular, finite difference based loss) adversary attacks by randomisation of the output of the network.

The idea appears to be somewhat novel considering that majority of existing methods consider randomize inputs or the model itself.

A natural question that would be particularly interesting to me is how does such defence compare against the defence by  randomizing the input and the model. There is no such comparison in the paper, which, to me, is the main weakness of the paper.

The authors consider the randomization in terms of Gaussian distribution. How would this differ if other types of distributions are considered, e.g. non-Gaussian distributions?

Section 4.2 considered the finite difference gradient error, and discussed the results for untargetted attacks. What bout targetted attacks?   The presentation of this section is also not very clear, e.g. the equation on page 6.

The citation style look odd to me, often you use either something like "[2,3]", or something like "Dhillon et al. (2018)", but not "(2,3)". In addition, the equations should be number for the convenience of references.

"... in our code \superscript {1}" - the links all point to other people's code, not your codes.

The notation for the output p is different from the notation in page 2 where you used y. Try to use consistent notations.

Regarding the novelty of this paper, I was based on my judgement and experience of reading a few papers, not I never published papers on adversary attacks or defence.

**Experience Assessment:**

I have read many papers in this area.

**Review Assessment: Checking Correctness Of Derivations And Theory:**

I assessed the sensibility of the derivations and theory.

**Review Assessment: Checking Correctness Of Experiments:**

I assessed the sensibility of the experiments.

**Review Assessment: Thoroughness In Paper Reading:**

I read the paper thoroughly.

---

> ### Author Response · Authors · 2019-11-14
> **Some clarification points**
>
> Thanks for your review, the following should address the points you raised.
> - We compare directly to the defense proposed in [26] in Table 2.
> - In the “Extensions” section of the Appendix, we considered noise sampled from a gaussian mixture model.
> - The analysis for targeted attacks is almost identical to untargeted attacks.
> - Our method involves a simple modification to the model under attack, and so our code changes to the attack code are very minimal.
> Thanks for pointing out errors in the notation and references.

---

### Official Review · AnonReviewer2 · 2019-10-28
**Official Blind Review #2**

**Rating:** 3

**Review:**

This paper proposes to introduce randomness in a classifier’s predictions to mitigate black-box attacks that rely on gradient estimation through finite differences. The intuition behind the defense is correct: finite differences rely on the outputs of the neural network being non-deterministic and accurate to estimate gradients near the test points being attacked.

However, the threat model chosen in this paper is not well justified: the adversary cannot be forced to use a particular strategy. Unlike what is suggested in Section 3, estimating gradients through finite differences is not the only strategy available in the black-box threat model (this is later mentioned in Section 6). In this case, the adversary could for instance decide to adapt by instead mounting a black-box attack that relies on transferability. Because Figure 2 shows that the defense does not provide robustness in the white-box setting, this suggests that other forms of black-box attacks that either (a) rely on transferability or (b) are label-based only would still evade the model. This limitation should be addressed to understand how applicable the defense strategy is in a realistic deployment.

Putting this aside, it is not clear from Figure 3 that an adaptive strategy was evaluated in the limited black-box setting that is considered here (the caption of Figure 3.b only describes a “white-box” adaptive adversary), or that the defense is effective. The attack success rates are high for many graphs and increase as the adversary averages over more runs. Moving forward, increasing further the highest number of runs would help appreciate the limitations of the approach: it is currently set to at most 100, which is low.

As far as organization is concerned, a lot of real estate is spent on background material, and few experimental results are presented to support claims made in the introduction. Addressing the above comments would probably require compressing background material a bit.

Page-by-page details:

1/ An attack is always adversarial by definition, “adversarial attack” is a tautology.

2 / What do you mean by “successful attacks”?

2/ What do you mean by “strongest” loss?

2/ Having a perturbation limited to be small does not guarantee it won’t impact the semantics of the input, even in the vision domain. Have you verified that the perturbations that you chose left semantics unperturbed?

2/ It is best to avoid making broad statements such as “We use the l2 perturbation penalty as this type of attack results in the strongest attacks” because they are unlikely to hold across datasets and models.

7/ Figures are difficult to parse (e.g., does T and U stand for targeted and untargeted?)

7/ Distillation was already shown to be vulnerable to black-box attacks in [7]

8/ Gradient masking was introduced in [7] prior to [25].

8/ It would be good to justify the following statement (see my comment above): “Although this is a valid attack vector for even black box models, we do not consider this type of attack in this work”

**Experience Assessment:**

I have published in this field for several years.

**Review Assessment: Checking Correctness Of Derivations And Theory:**

I assessed the sensibility of the derivations and theory.

**Review Assessment: Checking Correctness Of Experiments:**

I carefully checked the experiments.

**Review Assessment: Thoroughness In Paper Reading:**

I read the paper at least twice and used my best judgement in assessing the paper.

---

> ### Author Response · Authors · 2019-11-14
> **Why not transfer or white box - a rebuttal**
>
> The reviewer brings up a valid point that our proposed defense against finite difference black box attacks do not defend against white-box or transfer-based attacks. We agree with the reviewers on this point. Following the advice of [25] and to leave no room for assumptions on the effectiveness of the proposed defense against other attacks, we clearly defined our problem setting to be defense against finite difference attacks. [25] showed in 2018 that  7 out of 9 papers that proposed defense strategies in ICLR 2018 made false claims and over-promised their effectiveness. This includes defenses in our related work section, such as [26]. We found our proposed defense against finite difference attacks to be both novel and interesting, but we did not want to over-claim the defense's effectiveness. We are careful to only claim the defense is effective against (adaptive) finite different black box attacks.
>
> Indeed we evaluated an adaptive black box attacker, Figure 3b is describing a black box attacker that adopt the same adaptive strategy as the one in Figure 2b.
> In the black box case, we were limited by the available computational budget as increasing the number of samples in the adaptive case past 100 was very slow.
> Page by page responses
> 2- successful in the sense of satisfying the two conditions mentioned in the next sentence.
> 2- The authors in [18] consider the given loss function the best for their attack
> 2 – yes we verified the samples produced by the attacks were not semantically unperturbed in most cases
> 7- Correct, T=targeted and U=untargeted
> 7- The authors in [7] show defensive distillation is ineffective against a transfer attack, we show it is also ineffective against finite-difference based attacks. We believe the distinction is important and hope the reviewer agrees.
>
> [25] A. Athalye, N. Carlini, D. Wagner. "Obfuscated Gradients give a false sense of security: circumventing defenses to adversarial examples" ICML 2018
> [26] C. Xie, J. Wang, Z. Zhang, Z. Ren, and A. Yuille, “Mitigating Adversarial Effects Through Randomization” ICLR 2018

---

### Official Review · AnonReviewer4 · 2019-10-29
**Official Blind Review #4**

**Rating:** 3

**Review:**

This paper proposes applying randomization to the output layer of a DNN to defend against query-based attacks based on finite difference estimates. Then some theoretical analysis is provided, showing that with perturbation of a suitable scale, the randomization layer will not affect the accuracy of the model, while causing a large estimation error of finite difference methods that prevents finite-difference based attacks. Empirical results verify that the proposed defense is still effective against adaptive attacks where the randomness is averaged.

Pros:

The proposed method is simple, straightforward, yet novel. Its working mechanism is easy to understand and analyze, so it should be useful against finite-difference based attacks.

Limitation:

The proposed method is not useful to defense against white-box attacks and transfer-based attacks, since basically it does not change the predictive model. Some other randomization methods like [26], by contrast, change the predictive model, hence they may be useful against white-box attacks and transfer-based attacks.

Questions and suggestions:

This part is my main concern.

It seems that the experimental results are very good. For example, in Figure 3a, the defense is effective even if \sigma^2<1e-6. However, by the analysis in Section 4.2 (the formula below Line 3, Page 6), when \sigma^2=1e-6, |E[g_i-\gamma_i]| should be rather small, hence it should not block finite-difference based attacks. I think more explanation is needed for the good performance in the experiments.

Finite differences are extremely sensitive to small random perturbation of the function value when the spacing (step size) h is small. For example, g_i=\frac{L(f(x+he_i))-L(f(x-he_i))}{2h}, when h is very small, f(x+he_i) and f(x-he_i) is very close, hence adding perturbation to them will change g_i a lot. To present stronger adaptive attacks to output randomization, my suggestion is that a larger h can be adopted. It will be better if the results are investigated against attacks with different values of h.

A mistake:

In Section 4.1 on Page 4: "we can express the probability that x is misclassified in the vector d(p) as: \sum_{i=2}^C P(d(p_i)>d(p_m))". I think this is wrong, since P(A or B happens)=P(A happens)+P(B happens) only when A and B are mutually exclusive. However, "d(p_i)>d(p_m)" and "d(p_j)>d(p_m)" are not mutually exclusive. Hence, the probability that x is misclassified in the vector should be less than or equal to that sum of probabilities.

By the way, the writing in Section 4.1 is not clear:

- The overall misclassification probability is presented first, but after that K only represents the misclassification probability into a specific class. The connection between them is unclear.
- At the beginning of Section 4.1, the distribution of \epsilon is \epsilon\sim\mathcal{N}(\mu,\sigma^2\cdot\mathbf{I}_C): a unique \sigma is used. But after that, the variance of \epsilon_i becomes \sigma_i^2 instead of \sigma^2.
- In the second to the last line on Page 4, "level of noise (\sigma^2) can be set for each class separately", but the authors did not explain how to set them, and in the experiments \sigma^2 is set as the same scalar.
- In Figure 1b, the line style of "K=5.0e-3" and "K=1.0e-1" in the legend is very similar. The line style of "K=2.0e-01" in the legend is not clear: I do not know whether it refers to "-.-.-." or "------".

Typos:

Section 6, Page 8: "unintentionlly" => "unintentionally"
Some missing spaces after punctuation:
- Section 4.2, Page 5, "... gradient estimate.When the ..." => should add a space before "When"
- Section 5, Page 7, "In addition,input randomization ..." => should add a space before "input"

**Experience Assessment:**

I have published one or two papers in this area.

**Review Assessment: Checking Correctness Of Derivations And Theory:**

I carefully checked the derivations and theory.

**Review Assessment: Checking Correctness Of Experiments:**

I assessed the sensibility of the experiments.

**Review Assessment: Thoroughness In Paper Reading:**

I read the paper at least twice and used my best judgement in assessing the paper.

---

> ### Author Response · Authors · 2019-11-14
> **Adaptive attacker with large step size**
>
> Regarding the limitations wrt white box and transfer attacks, please see the first paragraph in our response to reviewer #2.
>
> Thanks for the suggestion for exploring the effect of increasing “h” as a type of adaptive attack. We missed this dimension as the three attacks we evaluated (especially [18]) recommended that “h” be set very small. As suggested by the reviewer, we ran the attacks with larger “h” (h=1e-3) and quickly realized why “h” must be small. When “h” is large, the success rate of the black box attack drops significantly.
> Thanks for pointing out the error in the probability, you are correct: the equal sign should be a <= according to the union bound.
> What we mean by “level of noise (\sigma^2) can be set for each class separately” is that \sigma_i for i = 2…C can be the same for each class (a spherical distribution) or different for each class based on the desired misclassification probability K as we show in section 4.1.
> The clarification comments and suggestions are also appreciated.

---

> > ### Comment · AnonReviewer4 · 2019-11-15
> > **The value of "h" and results of suggested adaptive attacks; A question not answered yet**
> >
> > Thanks for your reply!
> >
> > It seems that you did not mention the value of "h" used in the experiments in the paper (tell me if I am wrong). According to your code, it seems that h=1e-4 in the experiments. I think you should mention this in the paper.
> >
> > Thanks for your verification that when a larger "h" is used, the success rate of the black box attack drops significantly. It would be better if you could add the results with different "h" in the paper. According to my personal experience, a larger "h" such as h=1e-3 is still OK when attacking ordinary models although the performance would degrade a bit. Therefore it would be better if you present the experimental results carefully and clearly with a large "h" since it is likely to be a good adaptive attack.
> >
> > There is one remaining question in my review that you perhaps have missed: In Figure 3a, the defense is effective even if \sigma^2<1e-6. However, by the analysis in Section 4.2 (the formula below Line 3, Page 6), when \sigma^2=1e-6, |E[g_i-\gamma_i]| should be rather small, hence it should not block finite-difference based attacks. What is the reason? I guess that in theoretical analysis part, analyzing the bias is not the correct direction. Perhaps you should analyze the variance.

---

### Decision · Program_Chairs · 2019-12-19

**Decision:**

Reject

**Comment:**

This paper proposes a defense technique against query-based attacks based on randomization applied to a DNN's output layer. It further shows that for certain types of randomization, they can bound the probability of introducing errors by carefully setting distributional parameters. It has some valuable contributions; however, the rebuttal does not fully address the concerns.